# S. mansoni -derived omega-1 prevents OVA-specific allergic airway inflammation via hampering of cDC2 migration

Thiago A. Patente[1‡], Thomas A. Gasan[1‡], Maaike Scheenstra[1‡], Arifa Ozir-Fazalalikhan[1‡], Katja Obieglo[1], Sjoerd Schetters[2,3], Stijn Verwaerde[2,3], Karl Vergote[2,3], Frank Otto[1], Ruud H. P. Wilbers[4], Eline van Bloois[1], Yolanda van Wijck[5], Christian Taube[¤], Hamida Hammad[2,3,6], Arjen Schots[3], Bart Everts[1], Maria Yazdanbakhsh[1], Bruno Guigas[1], Cornelis H. Hokke[1], Hermelijn H. Smits[1] *

1 Department of Parasitology, Leiden University Center of Infectious Disease (LU-CID), Leiden University Medical Center (LUMC), Leiden, Netherlands, 2 Laboratory of Immunoregulation and Mucosal Immunology, VIB Center for Inflammation Research, Ghent, Belgium, 3 Department of Internal Medicine and Pediatrics, Ghent University, Ghent, Belgium, 4 Laboratory of Nematology, Plant Sciences Group, Wageningen University and Research, Wageningen, Netherlands, 5 Department of Pulmonology, LUMC, Leiden, Netherlands, 6 Department of Internal Medicine, Ghent University, Ghent, Belgium

¤ Current address: Department of Pulmonary Medicine, University Hospital Essen-Ruhrlandklinik, Essen, Germany
‡ TAP and TAG share first authorship on this work. MS and AO-F share second authorship on this work.
* h.h.smits@lumc.nl

**Data Availability Statement:** The authors confirm that all data underlying the findings are fully

## Abstract

Chronic infection with Schistosoma mansoni parasites is associated with reduced allergic sensitization in humans, while schistosome eggs protects against allergic airway inflammation (AAI) in mice. One of the main secretory/excretory molecules from schistosome eggs is the glycosylated T2-RNAse Omega-1 (ω1). We hypothesized that ω1 induces protection against AAI during infection. Peritoneal administration of ω1 prior to sensitization with Ovalbumin (OVA) reduced airway eosinophilia and pathology, and OVA-specific Th2 responses upon challenge, independent from changes in regulatory T cells. ω1 was taken up by monocyte-derived dendritic cells, mannose receptor (CD206)-positive conventional type 2 dendritic cells (CD206+ cDC2), and by recruited peritoneal macrophages. Additionally, ω1 impaired CCR7, F-actin, and costimulatory molecule expression on myeloid cells and cDC2 migration in and ex vivo, as evidenced by reduced OVA+ CD206+ cDC2 in the draining mediastinal lymph nodes (medLn) and retainment in the peritoneal cavity, while antigen processing and presentation in cDC2 were not affected by ω1 treatment. Importantly, RNAse mutant ω1 was unable to reduce AAI or affect DC migration, indicating that ω1 effects are dependent on its RNAse activity. Altogether, ω1 hampers migration of OVA+ cDC2 to the draining medLn in mice, elucidating how ω1 prevents allergic airway inflammation in the OVA/alum mouse model.

available without restriction. All relevant data are within the paper and its Supporting Information files.

**Funding:** The study was funded by research grants from the Lung Foundation Netherlands (3.2.10.072 and 5.1.15.015 to HS, https://research.longfonds.nl/subsidies) and Dutch Research Council (ZonMW-vidi: 91714352 to HS, https://www.zonmw.nl/nl/subsidie/nwo-talentprogramma-vidi-2023-0). The funders had no role in study design, data collection and analysis, decision to publish, or preparation of the manuscript.

**Competing interests:** HS receive research grants from the Lung foundation Netherlands and the Dutch Research Council and is a board member of the Netherlands Respiratory Society.

## Author summary

Asthma is a chronic inflammatory disease, leading to cough, wheeze, and shortness of breath. The prevalence has increased drastically in Westernized societies and is now increasing in low- and middle-income countries. Chronic infection with the parasitic helminth, *Schistosoma* (*S.*) *mansoni* protects against allergic airway inflammation (AAI) in mice, and is associated with reduced skin prick test positivity to inhaled allergens in humans. Here we show that peritoneal administration of ω1, a single glycoprotein secreted by *S. mansoni* eggs, reduced OVA/alum-induced AAI. ω1 is taken up in the peritoneal cavity by dendritic cells (DCs) expressing the mannose receptor, reducing their expression of CCR7 and migratory capacity to the draining mediastinal lymph nodes. This results in accumulation of DCs in the peritoneal cavity of allergic mice and reduced numbers of DCs reaching the draining lymph nodes. The effects observed for ω1 is dependent on its RNAse activity, since the RNAse mutant form of ω1 was unable to reduce AAI nor affect DC migration. Our findings provide insights into how ω1 can modulated the immune response during allergic inflammation, and this may open new avenues for the development of novel therapeutic strategies for allergic asthma.

## Introduction

Asthma affects more than 260 million people worldwide [1] and is characterized by a chronic inflammation in the lungs, leading to cough, wheeze, and shortness of breath. The prevalence has been rising since the second half of the previous century, firstly in Westernized countries [2] but now also in low- and middle-income countries [3]. It has been suggested that changes in lifestyle and microbial exposure, including helminth parasites, may play a role in the increased prevalence of allergies and possibly of childhood asthma [4,5]. Furthermore, chronic infection with *Schistosoma mansoni* [6,7], or with its eggs alone [8,9] can protect against OVA-induced allergic airway inflammation (AAI) in mice. Helminths, including *Schistosoma ssp.* express a large range of immunomodulatory molecules that affect host immune responses by inducing tolerance and exhaustion to prolong their survival in the host [10,11].

Omega-1 (ω1) is one of the most abundantly secreted proteins by mature *S. mansoni* eggs [12]. ω1 is a 31 kDa glycoprotein containing a $T_2$-RNAse domain [13], which mediates macrophage recruitment [14] and initiates granulomatous responses to eggs in the intestinal wall and liver [15]. In addition, ω1 induces Th2 polarization both in mice and humans [16,17], by modifying interactions between dendritic cells (DCs) and CD4[+] T lymphocytes. Indeed, human monocyte-derived DCs (moDCs) internalize ω1 via the mannose receptor [16,18] leading to increased IL-10 secretion [19], reduced CD86 expression and IL-12 secretion which is dependent on RNAse activity of ω1[16,18]. Additionally, ω1 stimulation impacts the cytoskeletal function of DCs, reducing membrane ruffling and limiting interaction with T cells [17]. Furthermore, ω1 induces FoxP3[+] regulatory T cells (Tregs) in NOD[+] mice and the expression of both CTLA-4 and IL-10 *in vivo* [19,20].

Although it has been shown that *S. mansoni* eggs can protect against AAI, the active molecules secreted by the eggs that mediate this protection are still poorly defined. Therefore, we here set out to address whether ω1 would be able to protect against AAI induction. Here, we show that peritoneal administration of recombinant ω1, but not the RNAse-defective mutant (Δω1), inhibits the development of OVA-induced AAI. This inhibition seems independent of changes in regulatory T cell populations, as those were not enhanced in the lungs or draining lymph nodes. To deduce the mechanism of ω1-induced protection against AAI, we examined

the cellular response to ω1 in peritoneal exudate cells (PEC) and ω1 effect on the processing and trafficking of OVA. We observed an accumulation of OVA$^+$ cDC2s in the PEC displaying decreased expression of activation markers and the chemokine receptor CCR7. This was associated with fewer OVA$^+$ cDC2 in the mediastinal lymph node (medLn) and reduced migratory capacity of PEC cDC2, explaining reduced allergen sensitization in these mice.

## Results

### Omega-1 administration reduces airway eosinophilia and OVA-specific Th2 priming

To investigate if the protection induced by *S. mansoni* egg against AAI could be attributed to the major secreted egg glycoprotein ω1, we used recombinant ω1, which has a similar effect as the native protein on human DCs [19]. The recombinant ω1 was compared with its mutant form (Δω1) in which RNAse activity was abolished through substitution of a histidine by a phenylalanine residue in its catalytic domain, leading to complete abolition of its RNAse capacity [19]. Mice were intraperitoneally injected with either ω1 or Δω1 twice (day -10 and -4) prior to allergic sensitization with OVA/alum, and followed by three OVA challenges one week later (**S1A Fig**). Compared to OVA/alum, ω1 significantly impaired overall cellularity and eosinophilia in bronchoalveolar lavage (BAL) (**Fig 1A**) and lungs (**Fig 1B**) of allergic mice. ω1 treatment reduced the numbers of DCs, neutrophils, T cells, and B cells in the BAL, as well as both CD11c$^+$ and CD11c$^-$ interstitial macrophages and a trend in neutrophils in the lungs (**Figs 1C and S2A**). OVA/alum-induced AAI reduced the numbers of alveolar macrophages in both BAL and lungs, and ω1 pre-treatment prevented this decrease in the BAL and the lungs (trend) of allergic mice. Importantly, most of the ω1-induced effects (e.g., lung eosinophilia (**Fig 1B**)) were either abolished or less potent in Δω1-treated mice, indicating that the RNAse activity is playing a key role. Additionally, ω1 administration (but not Δω1) reduced Th1- and Th2-associated cytokines in BAL fluid (**Fig 1D**) and OVA-specific Th2-associated cytokines in stimulated medLn cells (**Fig 1E**); contrary, ω1-specificTh2-associated cytokines were increased, confirming its prior described immune function (**Fig 1F**) [21]. Similar effects were also observed in the lungs, where Th2-associated cytokines (**Fig 1G**), but not CD8$^+$-secreted TNFα or IFNγ (**S1B Fig**) were reduced upon ω1 treatment. Even though, ω1 treatment did not seem to reduce serum levels of OVA-specific IgG1 or IgE, (**Fig 1H**), cellular infiltration around the airways was reduced by ω1, but not in Δω1-treated mice (**Fig 1I**). Next, we wanted to determine whether these cellular and immunological differences could also be associated with clinical outcomes. Lung resistance and elastance in OVA-induced AAI mice were significantly increased upon methacholine exposure, and pre-treatment with ω1 reduced both parameters. Interestingly, these differences were dependent on ω1 RNAse activity (**Fig 1J**). Taken together, these data suggests that ω1 administration protects mice against OVA-induced AAI, by a mechanism dependent of its RNAse activity.

### Lung cDC2 display a reduced activation profile during AAI in response to omega-1

To better understand how ω1 was protecting mice against OVA/alum-induced AAI, we further evaluated the cellular immune composition in lungs and medLn. Interestingly, different from the BAL and the lungs, ω1 administration did not decrease overall cellularity in medLn (**S1C Fig**). Additionally, it did not affect frequencies of total or CD44$^+$ effector CD4$^+$ T cells (**S1D Fig**). Furthermore, ω1 prevented the OVA/alum-induced reduction in FoxP3$^+$ T cells (**S1E Fig**) without affecting the expression levels of CTLA4 and GITR (**S1F Fig**). Because DCs

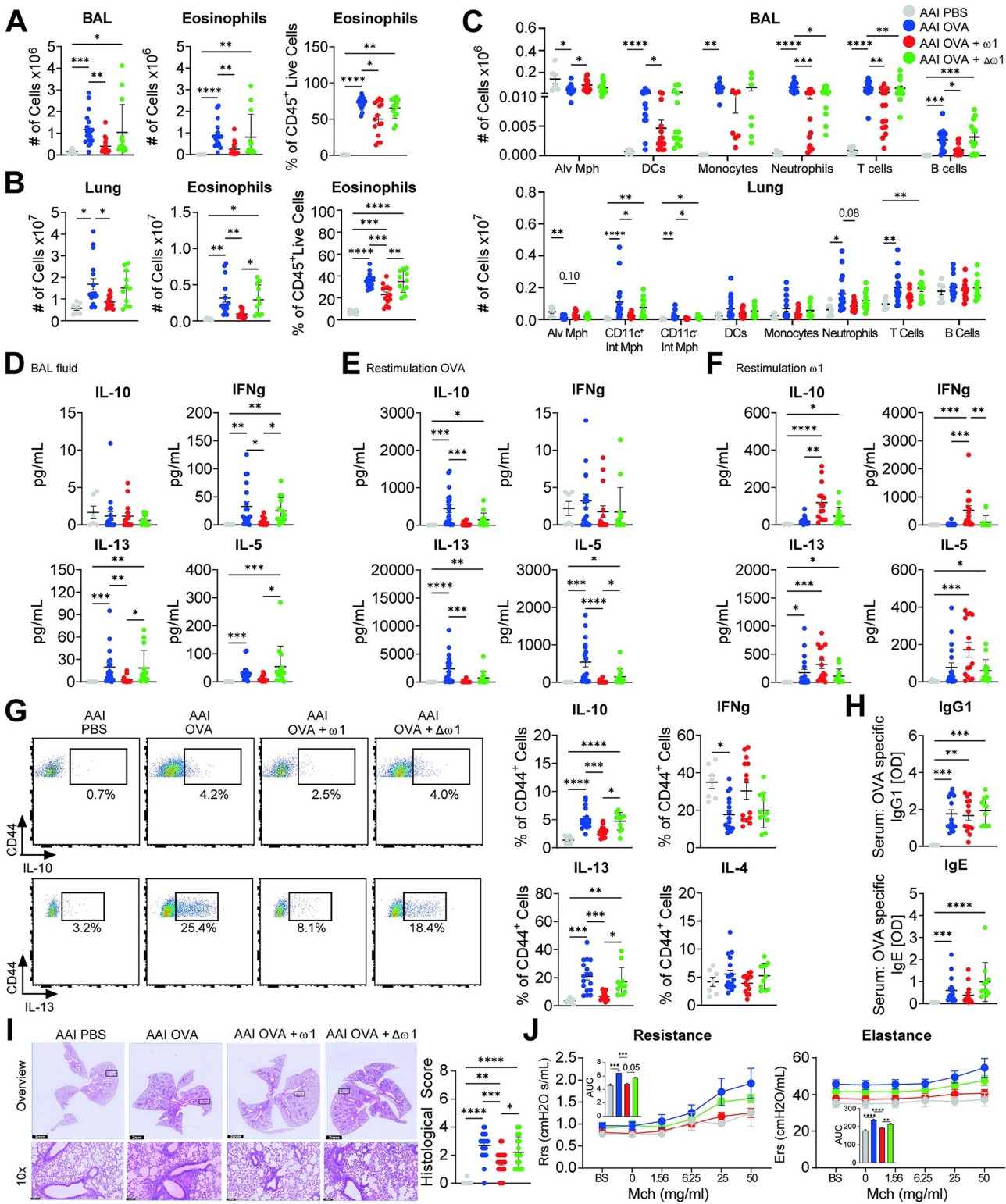

**Fig 1. Omega-1 pre-treatment reduces airway eosinophilia and OVA-specific Th2 priming.** Mice were pre-treated with 50 ug ω1 or Δω1 diluted in PBS on day 0 and 7. OVA sensitization was induced by 10 μg OVA intraperitoneal (i.p.) injected in alum adjuvant (2 mg/mL on day 12 and 19). Seven days after the last injection, mice were challenged for 3 consecutive days by intranasal (i.n.) administration of 50 μg OVA in PBS. Mice were sacrificed 24 hours after the last challenge. Total cell counts (left), number (middle) and frequency of eosinophils from bronchioalveolar lavage (BAL) **(A)** or lungs **(B)** of OVA/alum-induced allergic airway inflamed (AAI) mice. **(C)** Cell counts of different immune cells from BAL (upper) and lungs (bottom) of

allergic mice. **(D)** Cytokine concentration in BAL fluid measured by cytokine bead assay (CBA). Cytokine concentration in medLn cell supernatant after 4 days restimulation with OVA (10 μg/mL) **(E)** or with ω1 (10 μg/mL) **(F)** measured by CBA. **(G)** Frequency of cytokine-producing cells measured in the lungs of AAI mice by flow cytometry following polyclonal restimulation. **(H)** OVA-specific IgG1 and OVA-specific IgE antibodies in serum measured by ELISA. **(I)** Representative images of hematoxylin and eosin (H&E) staining from PFA-fixed lung secretions and scored for severity of cellular infiltration around the airways on a scale of 0–4. Scoring was performed blinded by two different individuals; the average of both scores is displayed. Bars = 100 μm. **(J)** Resistance and elastance measurements of lung function in response to increasing doses of methacholine, as assessed with the flexivent (SCIREQ). Data are pooled from 2 **(A-H)**, 3 **(I)** or 1 **(J)** independent experiments with 6 (PBS), 19 (AAI), 15 (AAI + ω1, and 14 (AAI + Δω1) mice **(A-H)**, with 9 (PBS), 26 (AAI), 22 (AAI + ω1, and 18 (AAI + Δω1) mice **(I)** or with 3 (PBS), 7 (AAI), 7 (AAI + ω1), and 5 (AAI + Δω1) mice **(J)**. Mean ± SEM are indicated in the graphs. One-way Anova with Tukey HSD post-test **(A and B frequency of cells;D-H;J)** or Kruskal-Wallys with Dunn's post-test **(A and B cell numbers;C)** was used to assess statistically significant differences; *p < 0.05, **p < 0.01, ***p < 0.001, ****p<0.0001.

are crucial in T cell priming and ω1 was demonstrated to affect T cell-priming capacity of DCs [16,18], we investigated whether ω1 administration affected lung DCs. Similar to mice injected with schistosome eggs [9], the frequency and cell numbers of lung moDCs were reduced in mice receiving ω1, however the mutant form did not restore moDCs composition (**S1G Fig**). Total lung CD11c$^+$ cDCs did not seem to be drastically modulated by ω1 administration (**Fig 1C**). We next performed an unbiased analysis on lung DCs, through dimensional reduction and unsupervised analysis flowcytometry data. We identified several phenotypic clusters that were altered in frequency by both OVA/alum and by ω1 pre-treatment (**Fig 2A**). Some of the cDC2 clusters (PG_05, PG_09, PG_10) were strongly increased in response to OVA/alum which was prevented by prior ω1 treatment (**Fig 2B and 2C**). These clusters, with PG_09 in particular, exhibited comparatively high levels of all activation markers, (**Figs 2D and S1H**). Conversely, two cDC1 clusters (PG_02, PG_16) were reduced by OVA/alum treatment and displayed a lower activation profile, while ω1 treatment prevented the reduction in those clusters (**Fig 2B, bottom right panel**).

Using manual gating we identified cDC1, cDC2 and a DC subset double negative for CD172a and XCR1 (DN DCs) (**Fig 2E**). Additionally, because ω1 can be internalized by the MR [18] we also evaluated the frequencies of cDC2 CD206$^-$ and cDC2 CD206$^+$. ω1 pre-treatment significantly prevented decreased frequencies of both cDC1 and cDC2 CD206$^+$ DCs in the lung induced by OVA/alum treatment (**Fig 2F**). Conversely, CD206$^-$ cDC2 and DN DCs were increased in OVA/alum mice, with ω1 significantly preventing the increase in the latter. Corroborating our unbiased analysis, OVA/alum increased CD80, CD86 and CD40 in all DCs subsets, while ω1 pre-treatment impaired CD40 expression in cDC2 and DN DCs subsets (**Figs 2G and S1J**). Surprisingly, Δω1 induced similar effects as ω1, suggesting that ω1-mediated inhibition of DC activation may occur independent of its RNAse activity. Taken together, the data suggests that ω1 administration prior to sensitization reduces OVA/alum-induced activation of lung cDC2s and increases the frequency of CD206$^+$ DCs independent of its RNAse activity.

## OVA uptake is increased and CCR7 expression is reduced on PEC CD206$^+$ cDC2 following omega-1 treatment

Next, we investigated the initial mechanisms by which ω1 induces protection against development of OVA/alum-induced AAI. For that, we evaluated the cellular composition of the PEC compartment (**S2B Fig**) and medLn (**S2C Fig**) 24 hours after the first injection of OVA/alum (**S3A Fig**). Pre-treatment with ω1, significantly increased total PEC cellularity, while its mutant form did not (**S3B Fig**). In agreement with previous work [22], OVA/alum induced a completed depletion of macrophages (**S3C Fig**), particularly resident macrophages (Res Mph) (**Fig 3A**), concomitant to an increase in eosinophils, neutrophils and DCs, both in cell numbers and frequency (**S3C Fig**). These observations were further enhanced by ω1 pre-treatment, but not by the mutant (**S3C Fig**). A more detailed analysis into the PEC DC subsets revealed

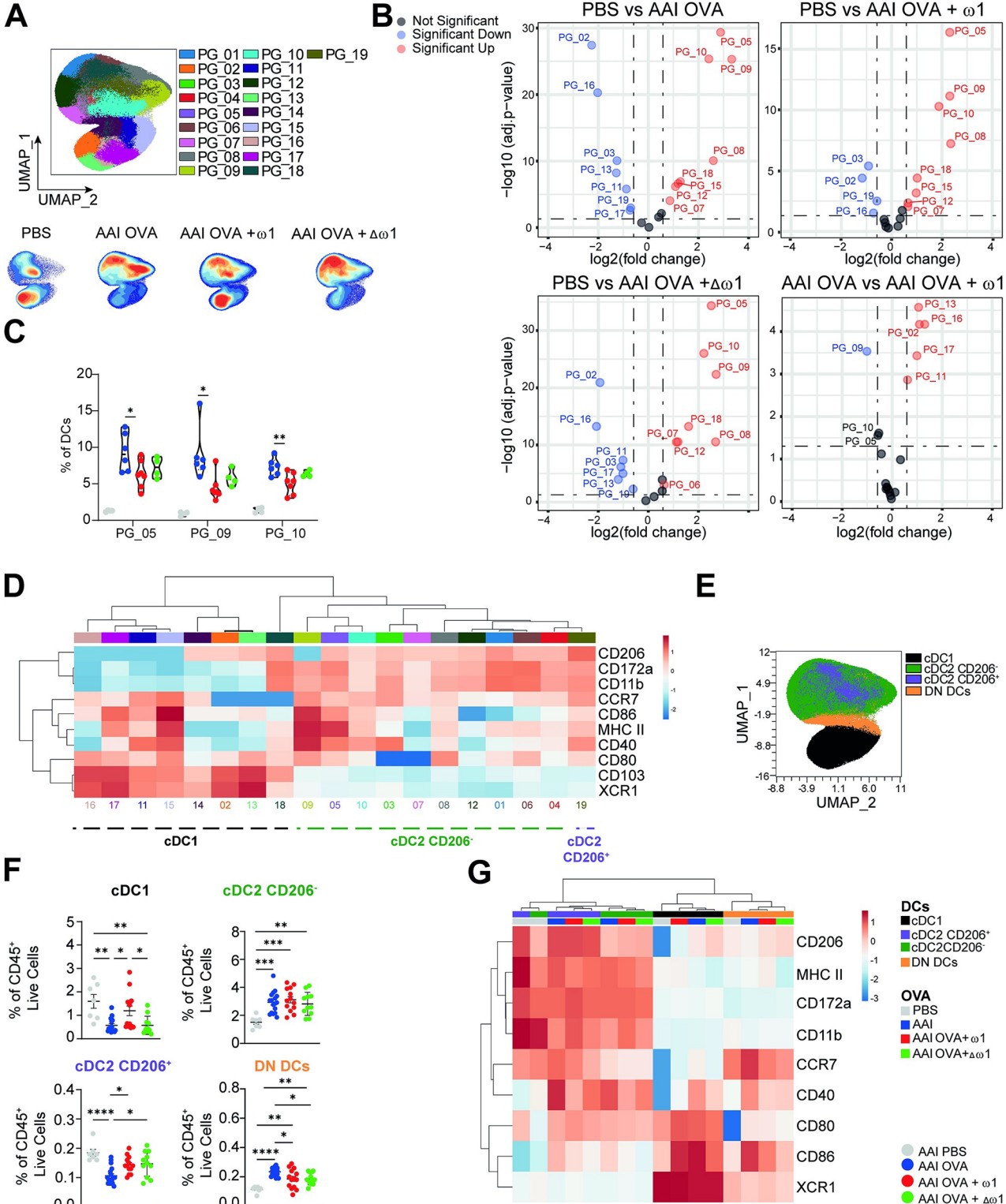

**Fig 2. Omega-1 pre-treatment increases lung cDC1 frequencies and reduces lung cDC2 activation.** Mice were treated as in Fig 1. **(A)** Phenograph clustering performed on lung DCs using activation and lineage markers (top) and contour plots overlaid on opt-SNE analysis displaying distribution of cells for each group (bottom). **(B)** Volcano plot displaying clusters as in **(A)** with significant differences in frequency between the groups. **(C)** Violins plots displaying frequencies of PG_05, PG_09 and PG_10 amongst the groups. **(D)** Heatmap of lineage and activation marker expression from lung DC clusters as defined in **(A)**. **(E)** Unbiased UMAP analysis of lung DCs populations in which cDC1, CD206⁺ cDC2, CD206⁻ cDC2 and DN DCs

are indicated. **(F)** Frequency of lung DCs subsets. **(G)** Heatmap of lineage and activation marker expression from lung DC subsets in AAI mice. Data are pooled from 2 independent experiments with 6 (PBS), 19 (AAI), 15 (AAI + ω1, and 14 (AAI + Δω1) mice **(F-G)** or representative of 1 out 2 independent experiments with 3 (PBS), 11 (AAI), 7 (AAI + ω1), and 7 (AAI + Δω1) mice **(A-E)**. Mean ± SEM are indicated in the graphs. One-way Anova with Tukey HSD post-test was used to assess statistically significant differences **(C** and **F)**, or EDGE-R analysis performed in OMIQ software with 1.5-fold change and adjusted p-value <0.05 considered as significant **(B)**; *p < 0.05, **p < 0.01, ***p < 0.001, ****p<0.0001.

profound changes in cDC1, cDC2 and DN DCs frequencies (**Figs 3B and S3E**) but not in moDCs (**S3D Fig**). To assess which cell type was taking up and processing OVA, we injected mice with either a fluorescent labelled OVA or with OVA-DQ (self-quenching conjugate exhibiting green fluorescence upon proteolytic degradation). Recruited macrophages (Rec Mph), moDCs and CD206$^+$ cDC2s were the primary cells taking up (**Fig 3C**) and processing OVA (**S3F Fig**), which was further enhanced upon ω1 pre-treatment (**Figs 3D and S3F**). Interestingly, in Δω1-pre-treated mice, uptake and processing of OVA by Rec Mph and CD206$^+$ cDC2s was not enhanced, suggesting that ω1 RNAse activity is required.

When evaluating activation markers, ω1 significantly prevented OVA/alum-induced CD80 and CD86 expression in both cDC1 and cDC2s while trends towards lower MHCII expression were also observed (**Fig 3E**). Intriguingly, the ω1 RNAse activity seems not involved after allergen challenge, similar to the observations in the BAL and lung tissue (**Fig 3E**). Because ω1 increased OVA processing by CD206$^+$ DCs, irrespective of lower MHCII expression, we hypothesized that ω1 induced accumulation of intracellular molecules of MHCII. However, we found that both ω1 and Δω1 decreased the level of intracellular MHCII, suggesting that ω1 does not induce intracellular accumulation of MHCII (**S3G Fig**). In fact, ω1 decreased intracellular levels of MHCII compared to PBS group, and this decrease was dependent on its RNAse activity, making it tempting to speculate that ω1 might prevent the synthesis of new MHCII molecules. An important feature of DCs is their migration from the tissue to draining lymph nodes to activate naïve T cells [23]. Previous data suggested that schistosome soluble egg antigen (SEA)-treated DCs were smaller and more mobile than immature DCs [24]. However, although we did not observe major differences in the size of DCs from ω1-treated compared to OVA/alum-treated mice (**S3H Fig**), ω1 immunization prevented OVA-alum induced filamentous actin (F-actin) expression (**Fig 3F**), with the exception of moDCs (**S3I Fig**), suggesting ω1 also alters DC cytoskeleton *in vivo*, as was previously reported *in vitro*. Furthermore, we found that both CD206$^+$ and CD206$^-$ cDC2s expressed higher CCR7 levels compared to other DC subsets and ω1 treatment significantly prevented the OVA/alum-induced CCR7 expression in cDC2s (**Fig 3G**), but not Rec Mph (**S3J Fig**), which was dependent on its RNAse activity. Interestingly, the levels of OVA inside OVA$^+$ cells, was not modulated by ω1 treatment (**S3K Fig**) despite increased frequencies of OVA$^+$ cells among CD206$^+$ cDC2s (**Fig 3D**). This, together with the reduced CCR7 expression in CD206$^+$ and CD206$^-$ cDC2s, might suggest that ω1 impairs the migration of these cells, leading to accumulation of OVA$^+$ cDC2s in the PEC. Of note, OVA$^+$ Rec Mph and moDCs also displayed increased OVA staining after ω1 treatment, suggesting a cell-nonspecific OVA accumulation (**S3J Fig**).

Finally, cytokine analysis showed that ω1, but not its mutant form, prevented OVA/alum-induced IL-1β secretion, but increased IFNγ, IL-10 and TNFα levels in the peritoneal cavity indicating that ω1 modulates the activity of local immune cells (**Fig 3H**). These differences, however, were less pronounced when cytokine levels were evaluated on a per-cell basis (**S3L Fig**), particularly for IFNγ levels and ω1-induced TNFα secretion (**S3M Fig**). Taken together, the data suggest that ω1 impairs the OVA/alum-induced activation, CCR7, and F-actin expression of PEC myeloid cells, particularly of cDC2s, leading to accumulation of OVA$^+$ cDC2s in the PEC, in a RNAse activity-dependent manner.

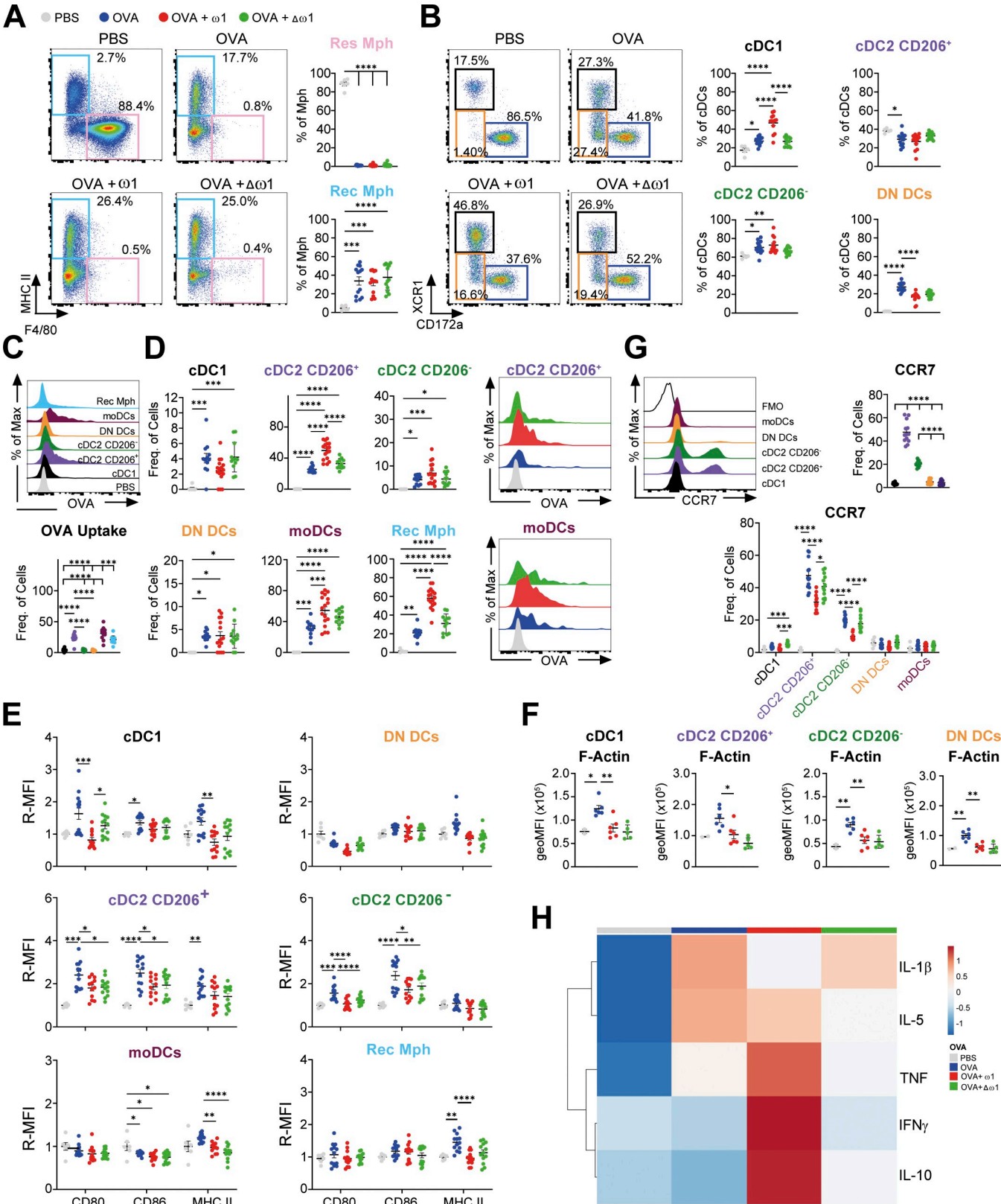

**Fig 3. Omega-1 increases OVA uptake in CD206+ cDC2 and reduces CCR7 expression.** Mice were treated as in in Fig 1, except that mice were sacrificed 24 hours after the first OVA/alum i.p. injection. **(A)** Representative dot plots (left) and frequency of resident (Res Mph) and recruited (Rec Mph) macrophages

(Mph) in peritoneal exudate cells (PEC) from OVA/alum sensitized mice. **(B)** Representative dot plots (left) and frequency of DC subsets in the PEC from OVA/alum sensitized mice. **(C)** Representative histogram of OVA uptake (upper) and frequency of OVA$^+$ cells in PEC of OVA/alum-injected mice (bottom). **(D)** Frequency of OVA$^+$ cells in PEC of OVA/alum-injected either treated or not with ω1 or Δω1. **(E)** Relative geometric MFI of the expression of CD80, CD86 and MHCII in the different DCs subsets and Rec Mph in PEC of OVA/alum-injected mice either treated or not with ω1 or Δω1. **(F)** geoMFI of phalloidin, as a readout for filamentous actin (F-Actin), in DCs from the PEC of OVA/alum treated mice. **(G)** Representative histogram of CCR7 (upper left), frequency of CCR7$^+$ cells in PEC of OVA/alum-injected mice (upper right) and frequency of CCR7$^+$ cells in OVA/alum-injected mice either treated or not with ω1 or Δω1 (bottom). **(H)** Heatmap displaying the average levels of cytokines in the PEC of OVA/alum-injected mice either treated or not with ω1 or Δω1. Data are from one experiment with 2 (PBS), 7 (OVA), 7 (OVA + ω1), and 6 (OVA + Δω1) mice **(F)** or pooled from 2 independent experiments **(A-E; G-H)** with 6 (PBS), 13 (AAI), 13 (OVA + ω1, and 13 (OVA + Δω1) mice **(A-E; G-H)**. Mean ± SEM are indicated in the graphs. One-way Anova with Tukey HSD post-test was used to assess statistically significant differences; *p < 0.05, **p < 0.01, ***p < 0.001, ****p<0.0001.

### PEC cDC2 from omega-1-injected mice display reduced migratory capacity

We next investigated DCs in the medLn that have migrated 24 hours after OVA/alum i.p. injection. Similar to the PEC, the medLn cellularity was increased by both ω1 and Δω1 (**S4A Fig**). Increased migratory MHCII$^{hi}$ cDCs (total migcDCs–numbers and trend for frequency) were observed in medLn of ω1-pre-treated mice (**S4B Fig**) but, differently from the PEC, this increase was driven by migcDC2s instead of migcDC1 (**Fig 4A**). The CD206$^+$ migcDC2s were the dominant DC subset taking up OVA (**Fig 4B**) and in stark contrast OVA$^+$ migDCs frequency (especially CD206$^+$ cDC2s) (**Fig 4C**) was reduced in the medLn of ω1-pre-treated mice, but not following pre-treatment with the mutant, suggesting that ω1 impaired the migration of DCs in an RNAse-dependent fashion. Importantly, the decrease in OVA$^+$ migDCs was also observed when cell numbers were calculated (**S4C Fig**), despite the increased numbers of (total) migDCs (**S4B Fig**). Interestingly, the expression of MHCII on all migcDCs, was significantly reduced in ω1-pre-treated mice, but not Δω1-pre-treated mice (**Fig 4D**). Furthermore, we observed increased F-actin levels in medLn migcDC1, but not in medLn migcDC2s of ω1-pre-treated mice (**S4C Fig**). Because ω1 pre-treatment impaired OVA/alum-induced CCR7 expression and F-actin in DCs from the PEC (**Figs 3G and S3H**), and the frequency of migrated OVA$^+$ DCs in medLn was reduced (**Fig 4C**), we hypothesized that ω1 hampers migration of activated DCs from the PEC. To test that hypothesis, we evaluated the migratory capacity of isolated PEC CD11c$^+$ DCs by a transwell-based migration assay *ex vivo*. After 24 hours, we determined the frequency of different DCs subsets within the migrated cells (bottom compartment). Indeed, cDC2 from ω1-pre-treated, but not from Δω1-pre-treated, mice displayed less migration *ex vivo* (**Fig 4E**). Collectively, the data show that ω1 pre-treatment reduces MHCII in medLn DCs and impairs the PEC cDC2 migration to medLns, possibly providing a mechanism through which ω1 induces protection against OVA/alum-induced AAI.

## Discussion

In the present study we show that intraperitoneally injected ω1 protects against AAI in an OVA/alum mouse model which we link to a hampered migration of OVA$^+$ cDC2 from the peritoneal cavity towards draining medLns through reduced expression of CCR7 and MHCII. Altogether, these data suggest that limiting DC migration is a viable approach to reduce allergen sensitization and may form a novel strategy for allergy treatment or prevention.

During *S. mansoni* infection schistosome eggs are potent drivers of Th2 polarization [25,16,17] and molecules secreted from mature eggs, such as ω1 and IPSE/α-1, induce the granulomatous responses indicative of chronic schistosomiasis [21,15]. Shifting the host responses towards a granulomatous, Th2 phenotype is beneficial to parasite survival by promoting egg excretion [26,14], preventing severe type I-induced immunopathology hindering egg release [27], and promoting host survival [11–13]. Conventional DCs are mainly responsible for presentation of egg antigens to CD4$^+$ T cells, and the polarization into effector T cell

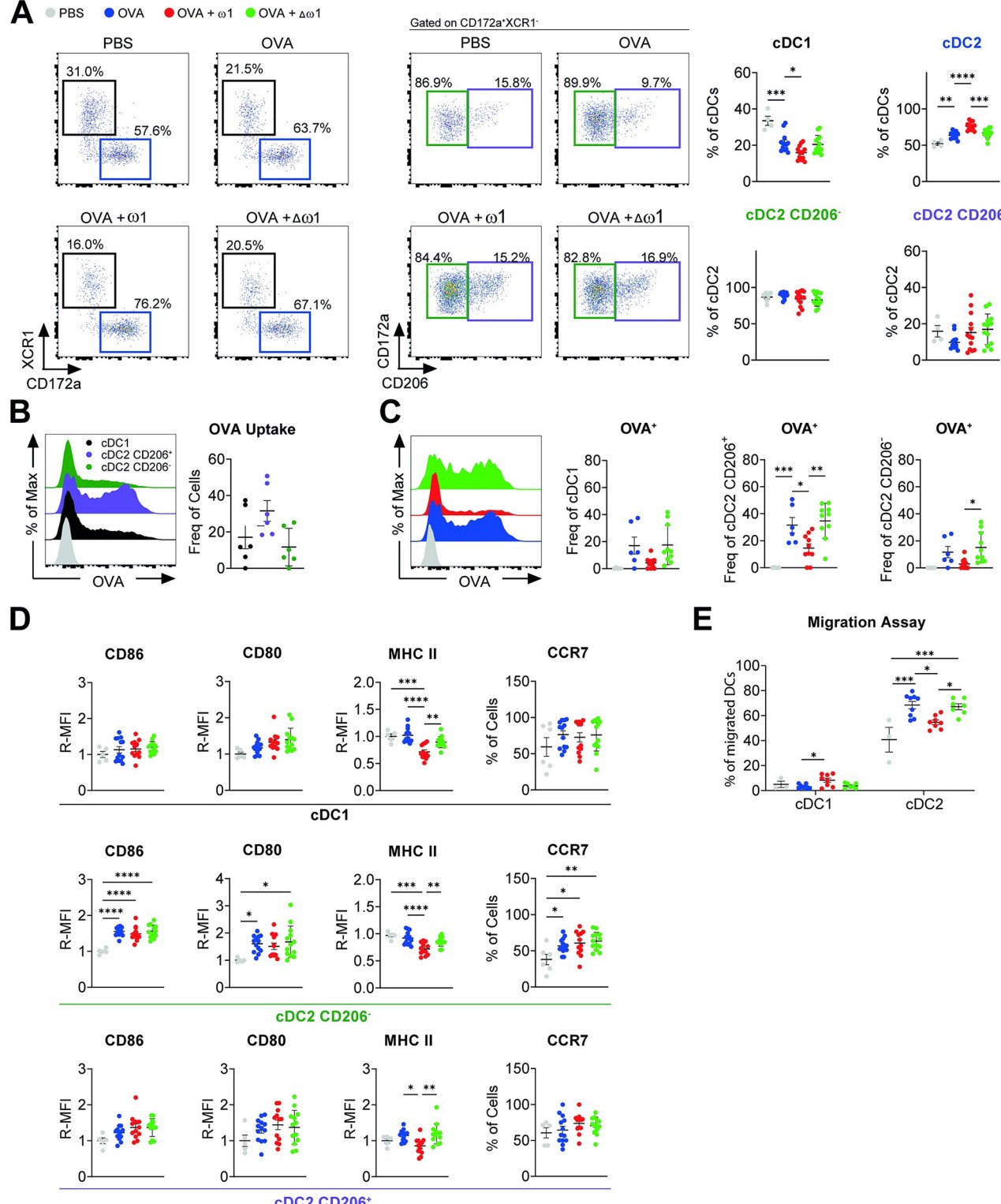

**Fig 4. Omega-1 reduces cDC2 migration.** Mice were treated as in Fig 3. **(A)** Representative dot plots of migratory cDC1 (migcDC1), migcDC2 (fair left), migcDC2 CD206+ and migcDC2 CD206- (middle) and frequency of migDC subsets in medLn of OVA/alum-injected mice either treated or not with ω1 or Δω1. **(B)** Representative histogram of OVA uptake (upper) and frequency of OVA+ cells in medLn of OVA/alum-injected mice (bottom). **(C)** Frequency of OVA+ migDCs subsets in medLn of OVA/alum-injected mice either treated or not with ω1 or Δω1. **(D)** Relative geometric MFI of the expression of CD80, CD86, MHCII and CCR7 in the different migDCs subsets in medLn of OVA/alum-injected mice either treated or not with ω1

or Δω1. **(E)** CD11c$^+$ cells were magnetically sorted from PEC of OVA/alum-injected mice either treated or not with ω1 or Δω1 and migratory capacity of both cDC1 and cDC2 was evaluated. Data are pooled from 2 independent experiments with 3 (PBS), 13 (AAI), 13 (AAI + ω1), and 13 (AAI + Δω1) mice **(A;D);** 6 (OVA) mice **(B);** 5 (PBS), 6 (OVA), 10 (OVA + ω1), and 10 (OVA + Δω1) mice **(C);** a pool of 3 independent experiments with 3 (PBS), 9 (OVA), 9 (OVA + ω1), and 9 (OVA + Δω1) mice per experiment that were pooled into groups of 3 mice per group before sorting of migDCs **(E)**. Mean ± SEM are indicated in the graphs. One-way Anova with Tukey HSD post-test was used to assess statistically significant differences; $^*$p $< 0.05$, $^{**}$p $< 0.01$, $^{***}$p $< 0.001$, $^{****}$p$<0.0001$.

responses [28]. During murine schistosomiasis cDC2s undergo a noticeable expansion in the liver and the draining lymph nodes [28,29]. Furthermore, cDC2 expansion is noticeable in the lung in both mice and humans, even before egg laying [30], probably due to the transit of immature parasites through the lung tissue. Despite this cellular expansion, populations from an endemic setting show a negative association between chronic schistosomiasis and allergic sensitization, with a reduced allergic Th2 response despite parasite-specific Th2 responses [31–33]. Furthermore, previous studies showed that, i.p. injected mature schistosome eggs could also protect against OVA-induced AAI in mice which coincided with the lack of a Th2 responses against OVA, despite the presence of SEA-specific Th2 responses [6,8,9]. Our study indicates that intraperitoneal administration of the *S. mansoni* glycoprotein ω1 is sufficient to replicate parasite-induced protection from AAI in the OVA-alum model. Also here, we observed reduced OVA-specific cytokines combined with ω1-specific Th2 cytokines upon restimulation with ω1. Although intraperitoneal administration of ω1 was shown to accumulate in abdominal organs and peritoneal draining lymph nodes [34] the distribution to the lungs seemed minimal. Nevertheless, we cannot exclude a direct effect of soluble transported ω1 on medLn cells explaining local ω1-specific Th2 cytokines. Because we observed less OVA-specific Th2 cytokines in ω1-treated mice, we suggest that lack of OVA-specific Th2 responses in the lung is due to reduced migration of OVA$^+$ cDC2 cells to medLn, rather than a direct effect of soluble transported OVA on medLn cells.

In order to engage its RNAse activity [13] and modulate the target cell, ω1 is primarily taken up through binding of its N-glycans [35] to mannose receptor (MR) CD206[18] which is expressed on most macrophage populations and DCs [36,37]. IL-4 and IL-13 can upregulate MR expression, while it is downregulated by IFN-γ [36]. In PEC, F4/80$^-$MHCII$^+$ cells, comprised of DCs, moDCs and Rec Mph, higher expression levels of CD206 are found compared to Res Mph [38]. Interestingly, we observed that following OVA/alum administration, the same populations (CD206$^+$ cDC2, moDCs and Rec Mph) were the primary myeloid cells taking up OVA, resulting in increased frequency of OVA$^+$ cells in the PEC of ω1-treated mice, after 24 hours. Interestingly, MR is also responsible for uptake of soluble (but not cell-associated) OVA in DCs [39]. These observations could be interpreted in several ways: ω1 increased OVA uptake; ω1 induced an accumulation of OVA$^+$ cells in the PEC or both. However, the increased frequency of OVA$^+$ DCs without significant changes in MFI suggests that the first option is unlikely and that DC subsets are rather retained in the PEC. This observation is supported by our findings that PEC DCs of ω1-treated mice expressed reduced CCR7 levels and exhibited less migration both *in* and *ex vivo*, as evidenced by reduced frequencies of OVA$^+$ migcDCs in medLns of ω1 treated mice. CCR7 expression was shown to be dependent on type 1 IFNs and was correlated with Th2 cell induction in a house dust mite (HDM) asthma model and during *S. mansoni* infections [40], suggesting a role for type 1 IFNs in ω1-reduced DC migration. Likewise, IL-10 was shown to reduce CCR7 expression and migration of DCs [41] and could be another mechanism explaining a lower migration of DCs, since we observed increased IL-10 levels in peritoneal lavage of ω1-treated mice. However, because both migration of DCs and IL-10 in peritoneal lavages were impaired in Δω1-treated mice, we cannot distinguish whether less CCR7 and DC migration are direct consequences of ω1 acting on DCs

(e.g., degrading CCR7 mRNA) or indirect consequences of ω1 acting on IL-10/type 1 IFNs-producing cells ultimately leading to more of those cytokines in the peritoneal cavity.

While ω1 did not affect OVA uptake by DC, it significantly increased processing of OVA (visualized by OVA-DQ transversion) by both CD206+ DCs and Rec Mph. Interestingly, we observed a negative correlation between MHCII expression and OVA+CD206+ DCs in the draining medLn 24 hours following OVA/alum injection. This correlation was inversed in the peritoneal cavity. The question remains whether increased OVA processing by CD206+ cDC2s correlates with increased OVA peptide bound to MHCII molecules on those cells. During the maturation of DCs, an increase in the biosynthesis of MHCII occurs to ensure efficient generation of peptide-MHCII complexes trafficking to the cell membrane [42]. Because ω1 did not increase intracellular MHCII levels, it seems unlikely that a defect in MHCII trafficking to the cell membrane explains reduced MHCII expression. Interestingly, while ω1 prevented the OVA-induced expression of MHCII on the surface levels, intracellular MHCII expression was reduced by ω1, and at least partially dependent on ω1 RNAse activity. This might suggest that ω1 mainly affects newly generated MHCII molecules by impairing the gene translation leading to a net reduction of MCHII levels. Due to overall decrease in MHCII and CCR7 expression, these cells would not be able to efficiently migrate and prime OVA-induced Type 2 responses, explaining the reduced allergen sensitization observed here following ω1 administration.

*In vitro* studies in human moDCs showed that both ω1 and SEA affect spreading and morphology of DCs [17,24]. Interestingly, we did not observe differences in cell size between ω1 treated and untreated DCs, but we did observe reduced F-actin staining in all DCs subsets, suggesting that ω1 can change DC structural morphology. It was shown that like LPS, SEA-treated DCs move farther and faster than immature DCs when tracking their movement on fibronectin-coated surfaces [24], but the effect of changes in CCR7 expression was not investigated in this context. Although we have demonstrated that DCs from ω1-treated mice showed an impaired migration in response to a chemokine gradient *in vitro*, we did not evaluate their ability to move or crawl and do not know whether this is affected or not. Whether ω1 itself has the same effect of SEA (a mixture in which ω1 is abundantly present) in terms of DCs motility, or whether SEA and potentially ω1 have the same effect *in vivo* as they have *in vitro*, is still subject for debate. F-actin is important for cell motility [43] and to the formation of the immunological synapse between DC and T cells [44], and ω1 has also been shown to impair the formation of DC-T cell conjugates [17]. Our observation that ω1-treated PEC DCs display reduced F-actin would be in line with the importance of F-actin to both migration and T cell priming by DCs. To what extend reduced CCR7 expression, reduced F-actin or a combination of both factors contributes to the reduced motility and/or reduced migration of PEC DCs is still an open question. Nevertheless, we argue that CCR7 expression may plays a more significant role in our model compared to F-actin. This is because the *in vitro* impaired migratory capacity of DCs and CCR7 expression both depended on ω1 RNAse activity, whereas the impaired F-actin formation observed in ω1-immunized mice did not. This suggests the reduced migratory capacity of PEC cDC2s could be associated with their reduced CCR7 expression rather than their reduced F-actin staining.

Intriguingly IL-1β secretion was reduced in the peritoneal cavity of ω1-treated mice 24 hours following OVA/alum injection, while TNFα levels were increased. This contrasts with earlier *in vitro* studies showing that both SEA and ω1 enhanced IL-1β secretion following TLR2 stimulation by peritoneal macrophages [45] and BMDCs [46], while TNFα levels were either unchanged or reduced, respectively. SEA-induced accumulation of IL-1β in the peritoneal macrophages was dependent on ω1 [45]. We here studied cytokine responses *in vivo* with different kinetics compared to the previously reported *in vitro* studies: we evaluated cytokines levels 24 hour following OVA/alum administration and 5 days after the last injection with ω1.

Furthermore, changes in cell composition and individual contributions of more than one cell type may explain different net results as compared to the *in vitro* studies: OVA/alum causes Res Mph depletion and DC recruitment to the peritoneal cavity by inducing uric acid [22], a well-known activator of inflammasome and IL-1β secretion [47]. In fact, moDCs showed enhanced IL-1β and IL-18 secretion when stimulated with alum and TLR-ligands [48]. Lastly, AAI induction in the absence of alum was shown dependent on both NLRP3 and IL1β expression in mice, and DCs from NLRP3$^{-/-}$ mice failed to migrate to draining medLns to prime immune responses [49]. This suggests that increased IL-1β levels in the peritoneal cavity is an important factor contributing to the induction of AAI. Therefore, we cannot rule out that ω1 might be preventing alum-induction of IL-1β secretion by macrophages and/or DCs and thus allergen sensitization and AAI. Additionally, we observed increased INF-γ both in the peritoneal cavity of ω1-pre-treated mice in the sensitization phase and by T cells in medLn of allergic mice after ω1 restimulation. The relationship between IFN-γ and allergic diseases has been suggested before [50], and systemic overexpression of IFN-γ prior to OVA/alum sensitization in mice, has been shown to supress AAI and DC migration [51]. Similarly, exposure to IFN-γ during pregnancy conferred protection against OVA-induced AAI in the adult life [52]. Therefore, it is tempting to speculate that the increased levels of IFN-γ observed in our model may be involved in the protective effect induced by omega-1. However, it is unclear during which phase of AAI (sensitization or challenge) IFN-γ would play a bigger role, nor whether the increase in IFN-γ is a cause or consequence of reduced Th2-associated T cells.

One of the limitations of the study is that we were not able to evaluate in depth the role of newly recruited monocytes to the peritoneal cavity in response to OVA/alum injections. It is established that OVA/alum depletes Res Mph from the peritoneal cavity [22] followed by replenishment of monocytes acquiring a F4/80$^{lo}$MHCII$^{+}$CSFR1$^{+}$CD11c$^{+}$ or a F4/80$^{lo}$MHCII$^{+}$CSFR1$^{+}$CD11c$^{-}$ profile [38]. These cells were described as macrophage precursors expressing high levels of RELMα and CD206. We were only partially able to evaluate the contribution of these precursors by studying the phenotype of F4/80$^{lo}$MHCII$^{+}$ cells, that would mirror the F4/80$^{lo}$MHCII$^{+}$CSFR1$^{+}$CD11c$^{-}$ population. We named them Rec Mph and we did not observe major differences in either the frequencies or the expression levels of CD80 and CD86; concordantly, CCR7 expression was very low and unchanged by ω1 in these cells. Because we did not have CSFR1 in our panels, we attempted to exclude CSFR1$^{+}$CD11c$^{+}$ precursors from DCs based on CD11b$^{+}$ expression. Even though we cannot completely exclude the possible contribution of these precursors to ω1-induced protection against AAI, we believe this is less likely because of the marginal effects of ω1 on this cell population.

The migratory capacity of DCs is an important feature of the DC biology, enabling efficiently T cell priming in the lymph nodes [53]. The amelioration of AAI has already been associated with impaired DC migration and CCR7 expression [54] suggesting that neutralizing DC lung chemotaxis could be beneficial for treatment against allergic diseases in patients [55,56]. An impaired CCR7 expression and DC migration has also been linked to amelioration of rheumatoid arthritis [57] and humanized mice treated with an anti-human CCR7 were completely resistant to collagen-induced arthritis [58]. Interestingly, FDA recently approved the first-in-human evaluation of an anti-hCCR7 mAb (targeting the ligand binding site of this receptor) in relapsing/refractory chronic lymphocytic leukaemia patients (CLL)[59,60]. This antibody was shown to bind to CCR7-expressing DCs, although not exclusively [59], and to impair the migratory capacity of cells obtained from CLL patients [60], opening the possibility to also exploit this signalling pathway in allergic patients.

In conclusion, our work demonstrates that ω1, through its RNAse activity, impairs the expression of costimulatory markers and CCR7 of (CD206$^{+}$) cDC2 in the peritoneal cavity, leading to a defect in their migratory capacity to draining lymph nodes which is associated

with protection of mice from OVA/alum-induced AAI. This may open new avenues for the development of novel therapeutic strategies for allergic asthma.

## Material and methods

### Ethics statement

Female wild-type C57Bl6/JOlaHsd mice, 7–12 weeks of age, were purchased from Envigo (Horst, The Netherlands). All mice were kept under specific pathogens free (SPF) conditions at the LUMC animal facility with free access to food and water under the guidelines for animal experimentation as approved by the Dutch 'Central Authority for Scientific Procedures on Animals' (CCD, License number: AVD1160020173525'and AVD11600202216417).

### Production of recombinant omega-1

Omega-1 (ω1—accession number ABB73003.1) and the H58F-omega-1 (Δω1—histidine residue (H) in the T2 RNAse catalytic domain was replaced by phenylalanine (F)) were produced in *Nicotiana benthamiana* plants as previously described [19]. The sequence of plant Δω1 and its RNAse activity has been compared with its native form previously, demonstrating that its capacity to degrade total liver RNA has been completely blunted [18].

### Induction of allergic airway inflammation (AAI)

Mice were pre-treated with 50 ug ω1 or Δω1 in PBS on day -11 and -4 by I.P. injection. OVA sensitization was induced by 10 μg OVA i.p. injection (Invivogen) in alum adjuvant (2 mg/mL; ThermoFisher Scientific) on day 0 and 7. Seven days after the last injection (day 14), mice were challenged for 3 consecutive days by intranasal exposure of 50 μg OVA in PBS. Mice were then sacrificed at day 17, 24 hours after the last intranasal challenge, and bronchial alveolar lavage (BAL), lungs, mdLns, and serum of mice were collected for further analysis. Lung function was performed using an invasive measurement of dynamic resistance (Flexivent, Scireq, Montreal, CA) as described previously [61].

### OVA sensitization protocol

Mice were pre-treated with 50 ug ω1 or Δω1 in PBS on day -11 and -4 by I.P. injection. OVA sensitization was induced by 10 μg OVA i.p. injection (Invivogen) in alum adjuvant (2 mg/mL; ThermoFisher Scientific) on day 0 and 7. Mice were then sacrificed at day 8, 24 hours after the last OVA i.p. intection, and peritoneal exudate cells (PEC), and mdLns of mice were collected for further analysis. Alexa Fluor 647-labeled OVA (self-conjugated with Alexa Fluor647 Conjugation Kit—Lightning-Link ab269823) or DQ ovalbumin (#D12053 –ThermoFisher) mixed with alum was administered at day 0 to detect OVA uptake or antigen processing, respectively.

### Tissue preparation

Bronchial alveolar lavage (BAL) and peritoneal exudate cells (PEC) were collected by flushing the lung or peritoneal cavity respectively with PBS-EDTA (2mM). Part of the lung was fixed by 3.7% PFA and used for histology. The other part of the lungs was minced in small parts using a scissor and enzymatically digested in complete RPMI media (cRPMI = RPMI-1640 supplemented with GlutaMAX, 5% heat-inactivated FCS [#S-FBS-EU-015, Serana, Pessin, Germany], 25 nM β-mercaptoethanol [#M6250, Sigma], 100 U/mL penicillin [#16128286, Eurecopharma, Ridderkerk, The Netherlands; purchased inside the LUMC] and 100 μg/mL streptomycin [#S9137, Sigma]) containing 2mg/mL of collagenase IV (Worthington) and 40 U/mL

DNase (Sigma-Aldrich). Tissues were digested for 45 min at 37˚C followed by a filtration step through a 100 μm strainer (#352360, BD Biosciences, Vianen, The Netherlands). Remaining red blood cells were lysed with 2 mL of red blood cell lysis buffer (RBC = 0.15M NH4Cl, 1 mM KHCO3, 0.1 mM Na2EDTA in ddH2O), for 5 min at room temperature before counting.

The spleen and the medLn were collected in 500 μL of no additives media (naRPMI = RPMI-1640 supplemented with GlutaMAX), in a plate and mechanically disrupted using the back-end of a syringe before addition of 50 μL of a digestion media (dRPMI = naRPMI supplemented with 11x collagenase D (#11088866001, Roche, Woerden, The Netherlands; end concentration of 1 mg/mL) and 11x DNase I (#D4263, Sigma, Zwijn-drecht, The Netherlands; end concentration of 2000 U/mL) for 20 minutes at 37˚C and 5% $CO_2$. Single cell suspensions were filtered after digestion with a 100 μm strainer (#352360, BD Biosciences, Vianen, The Netherlands) before counting in cRPMI. Spleens were subjected to red blood cell with RBC lysis buffer for 2 minutes at room temperature before counting. For serum collection, blood was collected via heart puncture, allowed to clog, spun down and serum was stored at -20˚C until further analysis.

## Flow cytometry

After processing, single cell suspensions from BAL, PEC, medLns and spleen were stained for 20 minutes at RT using the LIVE/DEAD Fixable Aqua Dead Cell Stain Kit (#L34957, Thermo) or LIVE/DEAD Fixable Blue Cells Stain Kit (#L34957, Thermo) and fixed for 15 minutes at RT using 1.85% formaldehyde (F1635, Sigma) before surface staining with antibodies in FACS buffer (PBS, 0.5% BSA [fraction V, #10735086001, Roche, Woerden, The Netherlands] and 2 mM EDTA) for 30 minutes at 4˚C. For the detection of intracellular and extracellular MHCII expression, cells were first stained with MHCII (2G9)–BUV 395 (BD #743876) before fixation. Then cells were permeabilized and stained with MHCII (2G9)–PE (BD #558593), washed and resuspended in FACS buffer for analysis at the flow cytometer. See S1 Table for further information on antibodies. For all staining, FcγR-binding inhibitor (2.4G2 Clone, kind gift of L. Boon, Bioceros), Brilliant Stain Buffer Plus (BD Bioscience # 566385) and True-Stain Mono-cyte Blocker (Biolegend—# 426103) was added. Flow cytometry was performed on FACS CANTO-II (BD), LSR-II (BD) or Cytek Aurora 5L (cytek bio) and analysed with FlowJo V10 (BD) and OMIQ (https://www.omiq.ai/).

## High dimensional spectral flow cytometry analysis

After tissue processing and single cell suspension isolation, lung cells were stained with a cock-tail of antibodies (S1 Table) and measured on a Cytek Aurora 5L. The raw fcs files were imported in OMIQ software and parameters were scaled using conversion factors ranging from 6000–20000. Samples were gated on live CD64⁻CD11c⁺MHCII⁺ lung DCs and subsam-pled using a maximum equal distribution across groups. After subsampling, Uniform mani-fold approximation and projection (UMAP) analysis was performed using lineage defining markers and activation markers as parameters in concatenated samples from all groups. Next, phenograph clustering (k = 100) was performed using the same parameters used for the UMAP. Data was further analysed with EdgeR to determine significant differences in the clus-ters among different genotypes and heatmaps and volcano plots were generated in R, using OMIQ-exported data for each cluster.

## Evaluation of CD11c⁺ cells migratory capacity

To evaluate the migratory capacity of PEC DCs, a transwell-based assay was performed with purified CD11c⁺ cells from the PEC. Cells were purified using magnetic beads (Mitenyi), and

$1x10^5$ cells were resuspended in serum free RPMI, and seeded in the membrane chamber of the CytoSelect Cell Migration Assay Kit (Cell Biolabs) containing polycarbonate membrane inserts (5 μm). 250 ng/mL CCL19 (TEBU-BIO BV Cat# P4614) diluted in 150 uL serum free RPMI or 150 uL of serum free RPMI was added in the lower chamber (feeder tray), and the membrane chamber was added in top of the feeder tray followed by incubation for 2 hrs at 37˚C. After the incubation period, the liquid present on the membrane chamber was carefully removed, and the membrane chamber was place in the Cell Harvesting Tray containing pre-warmed 150 uL of Cell Detachment Solution. The membrane chamber was incubated with Cell Detachment Solution for 20 min at 37˚C to detach migrated cells from the bottom part of the membrane. Next, cells that had migrated, and were present in the feeder tray, were pooled with the cells that were detached from the membrane, and stained with viability dye (see flow-cytometry section), MHC-II, CD11c, CD172a, SiglecH, XCR1, CD64, and CD45, acquired in a LSR-II (BD) and analysed with FlowJo V10 (BD).

## T cell restimulation

After isolation of lung immune cells, $1x10^6$ were platted in a 96 round bottom plate, and stimulated with 0.1 μg/mL PMA, 1 μg/mL ionomycin and 10 μg/mL BrefeldinA (all Sigma Aldrich) for 4 hrs at 37˚C. Next, cells were stained with viability dye and fixed with 1.85% formaldehyde (F1635, Sigma). Single cell suspensions were permeabilized eBioscience Permeabilization Buffer followed by intracellular staining with an antibody cocktail targeting T cells. After staining, cells were acquired in Cytek Aurora 5L (cytek bio) and analysed with FlowJo V10 (BD).

OVA- and ω1-specific responses were induced by 10 μg OVA or ω1 in $2x10^5$ medLns cells. After 96 hrs supernatants were collected for cytokine detection.

## Cytokine measurement by cytometric bead array

BAL fluid and cell culture supernatants were analysed for IFNγ, IL-5, IL-10 and IL-13 secretion using a cytokine bead array (CBA—#558298, #558302, #558300 and #558349 respectively and all BD Biosciences) on a flow cytometer as recommended by the manufacturer, but with both the beads and antibodies diluted 1:10 relative to the original recommendation.

## IgG1 and IgE measurement

Serum OVA-specific IgG1 and IgE antibodies were measured by ELISA, as previously described [9]. In brief, 25 μg/mL of OVA diluted in buffer (1M sodium carbonate) was used to coat a ninety-six-well Nunc Maxisorp plates (Thermo Fisher Scientific) at 4˚C overnight. After washing, wells were incubated with serum obtained from OVA-induced allergic mice, followed by incubation with biotinylated detection antibodies against IgG1 or IgE (BD Pharmingen), and horseradish peroxidase-conjugated streptavidin (BD Biosciences). Optical densities were measured at 450 nm after addition of TMB peroxidase substrate (KPL).

## Histology

Lungs were collected in 3.7% PFA and transferred after 1–2 days to 70% ethanol. Tissues were embedded in paraffin, 5 μm slices were cut and stained using hematoxylin and eosin (H&E; both Klinipath). Slices were blinded and scored 0–4 by two independent investigators using the Olympus BX41 light microscope (Olympus).

## Statistical analysis

Results are expressed as mean ± standard error mean (SEM) except stated otherwise. Continuous variables were log-transformed when not normally distributed (Shapiro-Wilk W test). Differences between groups were analysed by one way ANOVA or Kruskall-Wallis (non-parametric analysis) with Tukey HSD or Dunn's (non-parametric analysis) multiple comparison post-test as described in Fig legends. $p$ values $< 0.05$ were considered significant ($^*p < 0.05$, $^{**}p < 0.01$, $^{***}p < 0.001$) and statistical analyses were performed using GraphPad Prism v.9.3.

## Supporting information

**S1 Fig. Omega-1 reduced the frequency of moDCs in the lung of allergic mice.** Mice were pre-treated with either PBS, ω1, or mutant ω1 (Δω1) on day -10 and -4. OVA sensitization was induced by OVA/alum on day 0 and 7. Seven days after the last OVA sensitization, mice were challenged for 3 consecutive days by OVA aerosol exposure and sacrificed 24 hours after the last challenge. **(A)** Schematic view of OVA/alum-induced allergic airway inflammation, created with BioRender.com. **(B)** Frequency of TNFa$^+$CD44$^+$ (left) and IFNg$^+$CD44$^+$ effector CD8 T cells in the mediastinal lymph node (medLn) of allergic mice after restimulation with PMA and ionomycin in the presence of brefeldin A. **(C)** Total cell counts of medLn of allergic mice. **(D)** Frequency of total (left) and FoxP3$^-$CD44$^+$ effector (right) CD4 T cells in the medLn of allergic mice. **(E)** Representative FACS plot (left) and frequency of CD45$^+$ of regulatory FoxP3$^+$ T cells (right) in the medLn of allergic mice. **(F)** Representative histograms of CTLA-4 (top left) and GITR (bottom left) and relative geometric mean (R-MFI) of CTLA-4 (top right) and GITR (bottom right) of regulatory FoxP3$^+$ T cell present in the medLn of allergic mice. **(G)** frequency of CD45+ cells (left) and cell numbers (right) of monocyte-derived dendritic cells present in the lungs of allergic mice. **(H)** Uniform manifold approximation and projection (UMAP) projections of the lung dendritic cells compartment of allergic mice colored according to the expression of CD11b, XCR1, CD206, CD172a, CD40, CD86, CD80, and CCR7. **(I)** Cell numbers of DCs subsets identified in the lungs of allergic mice. **(J)** Geometric median fluorescent intensity (geoMFI) of the expression of CD80, CD86, MHC II, CCR7 and CD40 in lung DCs subsets of allergic mice. Data are pooled from 2 independent experiments with 6 (PBS), 19 (AAI), 15 (AAI + ω1, and 14 (AAI + Δω1) mice **(C-G; I)** or representative of 1 out 2 independent experiments with 3 (PBS), 11 (AAI), 7 (AAI + ω1, and 7 (AAI + Δω1) mice **(B;H; J)**. Mean ± SEM are indicated in the graphs. One-way Anova with Tukey HSD post-test was used to assess statistically significant differences **(B-G; J;I)**; $^*p < 0.05$, $^{**}p < 0.01$, $^{***}p < 0.001$, $^{****}p<0.0001$.
(TIF)

**S2 Fig. –Gating strategy for the identification of DCs and Mph across the different evaluated tissues.** Representative staining and gating strategies for **(A)** cDC1, CD206$^+$ and CD206$^-$ cDC2, double negative (DN) DCs, alveolar (Alv Mph) and interstitial macrophages (Int Mph) in the lungs; **(B)** cDC1, CD206$^+$ and CD206$^-$ cDC2, DN DCs, moDCs resident (Res Mph) and recruited macrophages (Rec Mph) in the peritoneal exudate cells (PEC); **(C)** cDC1, CD206$^+$ and CD206$^-$ cDC2, DN DCs and moDCs in the draining mediastinal lymph node (medLN).
(TIF)

**S3 Fig. Omega-1 induces increased processing but not uptake of OVA in CD206$^+$ cDC2s.** Mice were pre-treated with either PBS, ω1, or mutant ω1 (Δω1) at day 0 and day 7 and 4 days later were exposed to OVA/alum and sacrificed after 24 hours. Peritoneal exudate cells (PEC), lung and medLn were collected. **(A)** Schematic view of OVA/alum-AAI, created with BioRender.com. **(B)** Total cell counts of PEC from mice either pre-treated with ω1 or Δω1. **(C)**

Cell number (top panel) and frequency of live CD45$^+$ cell (bottom panel) from different immune cells in the PEC of OVA/alum-treated mice. **(D)** Frequency of CD45$^+$ cells from monocyte-derived dendritic cells (moDCs). **(E)** Representative FACS plot of CD206+ and CD206- cDC2 in the PEC of OVA/alum-treated mice. **(F)** Representative FACS plot (left) and quantification of OVA-DQ processing in the main myeloid cells in the PEC of OVA/alum-treated mice (middle chart) and in the PEC of mice treated with either ω1 or Δω1 (right chart). **(G)** Relative MHCII in the intracellular and extracellular compartment. **(H)** Relative MFI (R-MFI) normalized by PBS-treated mice for the quantification of DCs cell size using flow cytometer forward size scatter (FSC) as a parameter. **(I)** geoMFI of phalloidin, as a readout for filamentous actin (F-Actin), in moDCs and Rec MPh from the PEC of OVA/alum treated mice. **(J)** Frequency of CCR7-expressing recruited macrophages (Rec Mph) in OVA/alum treated mice. **(K)** Mice were injected with a mixture of OVA-Alexa Fluor 647 (AF647) and alum and 24 hours later cells from the peritoneal cavity were evaluated for the positivity of OVA. R-MFI of OVA-AF647, inside of OVA$^+$ cells, in DCs, moDCs and Rec Mph in the PEC of OVA-AF647/alum-treated mice. **(L)** Heatmap displaying the average levels of cytokines adjusted by cell counts in the PEC of OVA/alum-injected mice either treated or not with ω1 or Δω1. **(M)** Bar graphs of IL-10 and IFNg adjusted by cell counts in the PEC of OVA/alum-injected mice either treated or not with ω1 or Δω1 Data are from one experiment with 3 (PBS), 7 (OVA), 7 (OVA + ω1), and 7 (OVA + Δω1) mice **(F, I)**, pooled from 2 independent experiments with 6 (PBS), 13 (OVA), 13 (OVA + ω1), and 13 (OVA + Δω1) mice **(B-E; H; J-K),** or pooled from 2 independent experiments with 4 (PBS), 15 (OVA), 15 (OVA + ω1), and 12 (OVA + Δω1) mice **(G; L;M)**. Mean ± SEM are indicated in the graphs. One-way Anova with Tukey HSD post-test was used to assess statistically significant differences; *p < 0.05, **p < 0.01, ***p < 0.001, ****p<0.0001.
(TIF)

**S4 Fig. Omega-1 increases overall cellularity of draining medLN.** Mice were pre-treated with either PBS, ω1, or mutant ω1 (Δω1) at day 0 and day 7 and 4 days later were exposed to OVA/alum and sacrificed after 24 hours. PEC, lung and medLn were collected. **(A)** Total cell counts from mdLn of OVA/alum treated mice. **(B)** Cell number (top panel) and frequency of live CD45$^+$ cell (bottom panel) from different immune cells in the medLN of OVA/alum-treated mice. **(C)** geoMFI of phalloidin, as a readout for filamentous actin (F-Actin), in DCs from the medLN of OVA/alum treated mice. 2 independent experiments with 6 (PBS), 13 (OVA), 13 (OVA + ω1), and 13 (OVA + Δω1) mice **(A-B, D)** or 2 independent experiments with 7 (PBS), 10 (OVA), 10 (OVA + ω1), and 7 (OVA + Δω1) mice **(C)**. One-way Anova with Tukey HSD post-test was used to assess statistically significant differences **(A-D)**; *p < 0.05, **p < 0.01, ***p < 0.001, ****p<0.0001.
(TIF)

**S1 Table. Antibody list.** List of antibodies used in the current manuscript.
(DOCX)

**S1 Data. Data availability.** Values used to generate the graphics displayed in all figures in the current manuscript.
(XLSX)

# Author Contributions

**Conceptualization:** Thiago A. Patente, Thomas A. Gasan, Maaike Scheenstra, Christian Taube, Hamida Hammad, Bart Everts, Maria Yazdanbakhsh, Bruno Guigas, Cornelis H. Hokke, Hermelijn H. Smits.

**Formal analysis:** Thiago A. Patente, Thomas A. Gasan, Maaike Scheenstra, Katja Obieglo, Eline van Bloois.

**Funding acquisition:** Hermelijn H. Smits.

**Investigation:** Thomas A. Gasan, Maaike Scheenstra, Arifa Ozir-Fazalalikhan, Katja Obieglo, Sjoerd Schetters, Stijn Verwaerde, Karl Vergote, Frank Otto, Eline van Bloois, Yolanda van Wijck.

**Methodology:** Thiago A. Patente, Thomas A. Gasan, Maaike Scheenstra, Arifa Ozir-Fazalalikhan, Katja Obieglo, Sjoerd Schetters, Stijn Verwaerde, Karl Vergote, Yolanda van Wijck.

**Project administration:** Hermelijn H. Smits.

**Resources:** Ruud H. P. Wilbers, Hamida Hammad, Arjen Schots.

**Supervision:** Christian Taube, Hamida Hammad, Bart Everts, Maria Yazdanbakhsh, Bruno Guigas, Cornelis H. Hokke, Hermelijn H. Smits.

**Writing – original draft:** Thiago A. Patente, Thomas A. Gasan, Maaike Scheenstra, Hermelijn H. Smits.

**Writing – review & editing:** Thiago A. Patente, Thomas A. Gasan, Maaike Scheenstra, Arifa Ozir-Fazalalikhan, Katja Obieglo, Sjoerd Schetters, Stijn Verwaerde, Karl Vergote, Frank Otto, Ruud H. P. Wilbers, Eline van Bloois, Yolanda van Wijck, Christian Taube, Hamida Hammad, Arjen Schots, Bart Everts, Maria Yazdanbakhsh, Bruno Guigas, Cornelis H. Hokke, Hermelijn H. Smits.

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
