## [Decision Letter · Decision Letter 0]

5 Oct 2023

Dear Dr. Smits,

Thank you very much for submitting your manuscript "S. mansoni -derived omega-1 prevents OVA-specific allergic airway inflammation via hampering of cDC2 migration." for consideration at PLOS Pathogens. As with all papers reviewed by the journal, your manuscript was reviewed by members of the editorial board and by several independent reviewers. In light of the reviews (below this email), we would like to invite the resubmission of a significantly-revised version that takes into account the reviewers' comments.

We cannot make any decision about publication until we have seen the revised manuscript and your response to the reviewers' comments. Your revised manuscript is also likely to be sent to reviewers for further evaluation.

Sincerely,

Michael H. Hsieh

Guest Editor

PLOS Pathogens

Margaret Phillips

Section Editor

PLOS Pathogens

Kasturi Haldar

Editor-in-Chief

PLOS Pathogens

orcid.org/0000-0001-5065-158X

Michael Malim

Editor-in-Chief

PLOS Pathogens

orcid.org/0000-0002-7699-2064

Reviewer's Responses to Questions

**Part I - Summary**

Reviewer #1: The manuscript number PPATHOGENS-D-23-01084 reported the host immune response from one of the major proteins from mature Schistosoma mansoni egg secretory product (ESP) named omega-1 to prevent OVA-specific allergic airway inflammation. The authors used either wild type and mutant omega-1 (lack of its ribonuclease activity) recombinant proteins to induce mouse immune cells in vivo prior to expose to OVA immunogen, followed by investigating of secretory cytokines in circulation, intersted tissue immune cell populations and their secretions in vitro. The concept of study and the outcome would be great and fullfill the knowledage about this protein which have been reported as important ESP to egg-induced granulomas in mammalian host. However, the current manuscript has not reached the satisfaction to be accepted for publication yet. The manuscript would be suitable for publications after authors provide more information in the manuscript text especially the results, materials, and method sections. The complexity of experimental setting staring from mouse immune induction by wild type or mutant omega-1 proteins, I would recomment to add the schematic of the experiments cluding mouse immune IP, tissues, and blood sample collection til the down stream experiment ex vivo/in vitro. I am strontly to improve the writing to clarify unclear text especially in Results and Materials and Method sections.

Reviewer #2: This is a complex study (depicted in 4 dense Figures) which focuses on understanding how a major schistosome egg protein, Omega1, suppresses the allergic airway inflammation (AAI). Experimentally it is a strong study, but the data are poorly presented and described. There are too many turns and twist with unnecessary data that asks from a reader to decipher which information is important and which is not.

**Part II – Major Issues: Key Experiments Required for Acceptance**

Reviewer #1: Improve the writing to clarify the result, materials and methods and figure legend sections.

Reviewer #2: This is a complex study (depicted in 4 dense Figures) which focuses on understanding how a major schistosome egg protein, Omega1, suppresses the allergic airway inflammation (AAI).

Required Experiments

While the authors analyzed several cellular parameters as read-out of AAI, I believe that a major one is missing: a functional test of lung capacity (e.g. Penh measurements 24 hours after the final airway challenge using a Buxco system with mice exposed to increasing doses of methacholine or a similar technique). This would be important before proposing a new therapeutic approach based on the observations made in this manuscript.

The observation that immunization (pre-treatment) with Omega1 triggers a significant IFN-gamma response was not explored by the authors as a possible underlying mechanism of the immunomodulatory effect of Omega1 on AAI, especially since the IFN-gamma response was dependent on the intact RNAse-activity of Omega1, which could explain decreased eosinophil recruitment in lungs of OVA/alum-primed animals.

The authors should address the role of IFN-gamma induced by Omega-1 as a factor that mediates suppression of AAI.

**Part III – Minor Issues: Editorial and Data Presentation Modifications**

Reviewer #1: o Result section is lack of matric and statistic information in manuscript text. This section needs to be improved to clarify allf of the matric data of individual panel of each figure, along with the value of statistic analysis and number of samples used for statistic analysis.

- Figure legends are insufficient information of the explanation of all the graph such as x- and y-axis, color code used especiall heat maps

- No scale bar(s) in panel I of figure 1.

o Material and Methods section

- Consider adding the schematic of the experimental set up, since there were complex experimental study staring from mouse immune induction through the ex vivo/in vitro experiments

- There was several unclear information as list below, and lack of references of techniques used in this study.

• Lack of H58F-omega-1 sequence information e.g. GenBank accession number or other public sequence submission

• First used abbreviations were not clarified, for example, SPF (line 331), LUMC (line 331), OVA (line 334), medLn (line 344), UMAP (line 367).

• ‘in vivo expeiments’ is confusion, the author indicated that mice were pre-treated with omega-1 protein, but there is no explanation of how the authors proceeded for pre-treatment. Was it IP injection? The time of IP injection of OVA and mouse sacrification is unclear.

• ‘Tissue preparation’; the methodology of this sub-section must be provided. It is unclear that after tissue enzymatic digestion, how blood cells need to be lysed (remaining blood cells on lines 346-247). Was it meant collagenase fail to complete to lyse the blood cells? Did tissues mince before enzymatic digestion?

• ‘High dimentional spectral flow cytometry analysis’; what kind of samples (line 365) used in this study? Did they the raw data from flow cytometry?

• ‘Migration’; what was migration? Cell migration?, What kind of the medium used(line 376)? What was the diluent of CCI19? What is the detact buffer? Please clarify feeder tray and detach tray and strongly to include the reference(s).

• ‘T cell restimulation’; it is unclear what kind of cell(s) used in this study. Only Tcell or both DC and Tcell? Please provide more information including reference(s)

• ‘cytokine maesurment’; authors indicated that the IL-5, IL-10, IL-13 and IFN-g concentration were determined, followed by flow cytometry. Did author meant those secreted cytokines measured from serum and/or tissue culture medium? How flow cytometry used for cytokine measurement?

• ‘IgG1 and IgE measurement’; did authors meant sandwich ELISA that enzymatic plate coated with IgG and IgE, the probed with OVA, then react with its antibody before conjugated-antibody kick in? How these antibodies specific to OVA? Antibody enrichment? Please clarify.

Reviewer #2: Fig. 2 describes the analysis of lung DC (Omega1 pre-treatment increases lung cDC1 frequencies and reduces lung cDC2 activation), although the effects observed are independent of RNAse activity of Omega1 and, as such, appears to be less relevant to this study because Omega1-mediated suppression of AAI is dependent on its RNAse activity. Indeed, the results in this figure are not even mentioned in the Abstract.

In several instances, a more rigorous interpretation of the data may be required: E.g. “except BAL macrophages and lung alveolar macrophages, that were reduced in OVA/alum mice but increased in ω1-treated mice (Fig. 1C)” (line 82). However, BAL macrophages in omega1-treated mice were also significantly and dramatically decreased, but just slightly less than in the only OVA/alum-treated group. Moreover, there are no significant changes in lung cell composition based on Fig 1B, except for a decreased number of eosinophils (as stated in Abstract).

It is not clear how the authors reconcile their description of Omega1 as a “general adjuvant” (Line 88), based on induction of Th2 cytokine by Omega1 in mice that were immunized only with OVA and its immunosuppressive effect of Omega1 on OVA-induced AAI.

Focusing on very early events after OVA/alum immunization (day 1 after 1st injection) at the site of injection and draining mediastinal nodes (mLN), authors show that the frequency of cDC2 that have picked-up OVA is lower in animals treated with Omega1. However, since they also show a significant increase in the number of cDC2 in the same animals, the authors need to show the absolute number of OVA+ cDC2 in different experimental groups to make a convincing statement.

As a punch line, the authors perform a migratory assay showing that cDC2 from Omega1-pretreated mice display (~20%) decreased ability to migrate in an in vitro assay in response CCL19. The degree to which decreased DC expression of CCR7 or F-actin (need to be brought up in Abstract and main Figure) in mice primed with Omega1 contributes to “slower” migration is unclear at present.

PLOS authors have the option to publish the peer review history of their article (what does this mean?). If published, this will include your full peer review and any attached files.

Reviewer #1: No

Reviewer #2: No
---

## [Editor Report · Decision Letter 1]

27 Jul 2024

Dear Dr. Smits,

We are pleased to inform you that your manuscript 'S. mansoni -derived omega-1 prevents OVA-specific allergic airway inflammation via hampering of cDC2 migration.' has been provisionally accepted for publication in PLOS Pathogens.

Best regards,

James J Collins III

Section Editor

PLOS Pathogens

James Collins III

Section Editor

PLOS Pathogens

Michael Malim

Editor-in-Chief

PLOS Pathogens

orcid.org/0000-0002-7699-2064
---

## [Editor Report · Acceptance letter]

12 Aug 2024

Dear Dr. Smits,

We are delighted to inform you that your manuscript, "S. mansoni -derived omega-1 prevents OVA-specific allergic airway inflammation via hampering of cDC2 migration.," has been formally accepted for publication in PLOS Pathogens.

Best regards,

Michael Malim

Editor-in-Chief

PLOS Pathogens

orcid.org/0000-0002-7699-2064